# Multivalvular Endocarditis: A Rare Condition with Poor Prognosis

**DOI:** 10.3390/jcm11164736

**Published:** 2022-08-13

**Authors:** Sara Álvarez-Zaballos, Victor González-Ramallo, Eduard Quintana, Patricia Muñoz, Sofía de la Villa-Martínez, M. Carmen Fariñas, Francisco Arnáiz-de las Revillas, Arístides de Alarcón, M. Ángeles Rodríguez-Esteban, José M. Miró, Miguel Angel Goenaga, Josune Goikoetxea-Agirre, Elisa García-Vázquez, Lucía Boix-Palop, Manuel Martínez-Sellés

**Affiliations:** 1Cardiology Department, Hospital General Universitario Gregorio Marañón, 28027 Madrid, Spain; 2Home Hospitalization Department, Hospital General Universitario Gregorio Marañón, 28027 Madrid, Spain; 3Cardiovascular Surgery Department, Hospital Clinic, 08036 Barcelona, Spain; 4Clinical Microbiology and Infectious Diseases Department, Hospital General Universitario Gregorio Marañón, Health Research Institute Gregorio Marañón, 28027 Madrid, Spain; 5CIBER Enfermedades Respiratorias-CIBERES (CB06/06/0058), School of Medicine, Universidad Complutense de Madrid, 28040 Madrid, Spain; 6Infectious Diseases Department, Hospital Universitario Marqués de Valdecilla, IDIVAL, Universidad de Cantabria, 39008 Santander, Spain; 7CIBER de Enfermedades Infecciosas-CIBERINFEC (CB21/13/00068), Instituto de Salud Carlos III, 28029 Madrid, Spain; 8Clinical Unit of Infectious Diseases and Clinical Microbiology, Infectious Diseases Research Group, Institute of Biomedicine of Seville (IBIS), University of Seville/CSIC/University Virgen del Rocío, 41013 Seville, Spain; 9Cardiovascular Surgery ICU Department, Hospital Central de Asturias, 33011 Oviedo, Spain; 10Infectious Diseases Service, Hospital Clínic—IDIBAPS, University of Barcelona, 08036 Barcelona, Spain; 11Infectious Diseases Department, Hospital Universitario Donosti, 20014 San Sebastián, Spain; 12Infectious Diseases Department, Hospital Universitario de Cruces, 48903 Bilbao, Spain; 13Internal Medicine-Infectious Diseases Department, IMIB, Hospital Clínico Universitario Virgen de la Arrixaca, Facultad de Medicina, Universidad de Murcia, 30120 Murcia, Spain; 14Infectious Diseases Department, Hospital Universitari Mútua Terrassa, 08221 Barcelona, Spain; 15CIBERCV (CIBER Enfermedades Cardiovasculares), Universidad Europea, Universidad Complutense, 28040 Madrid, Spain

**Keywords:** infective endocarditis, multivalvular endocarditis, prognosis, mortality

## Abstract

Background. Infective endocarditis (IE) is a severe condition. Our aim was to describe the profile and prognosis of patients with multivalvular infective endocarditis (MIE) and compare them to single-valve IE (SIE). Methods. We used a retrospective analysis of the Spanish IE Registry (2008–2020). Results. From 4064 definite cases of valvular IE, 577 (14.2%) had MIE. In patients with MIE, the most common locations were mitral (552, 95.7%) and aortic (550, 95.3%), with mitral-aortic involvement present in 507 patients (87.9%). The most common etiologies were *S. viridans* (192, 33.3%) and *S. aureus* (113, 19.6%). MIE involved only native valves in 450 patients (78.0%). Compared with patients with SIE, patients with MIE had a similar age (69 vs. 67 years, respectively, *p* = 0.27) and similar baseline characteristics, but were more frequently men (67.1% vs. 72.9%, *p* = 0.005) and had a higher incidence of intracardiac complications (36.2% vs. 50.4%, *p* < 0.001), heart failure (42.7% vs. 52.9%, *p* < 0.001), surgical indication (67.7 vs. 85.1%, *p* < 0.001), surgery (46.3% vs. 56.3%), and in-hospital mortality (26.9% vs. 34.3%, *p* < 0.001). MIE was an independent predictor of in-hospital mortality (odds ratio (OR) 1.3, 95% confidence interval (CI) 1.1–1.7, *p* = 0.004) but did not have an independent association with 1-year mortality (OR 1.1, 95% CI 0.9–1.4, *p* = 0.43). Conclusions. About one-seventh of the valvular IE patients had MIE, mainly due to mitral-aortic involvement. MIE is associated with a poor in-hospital prognosis. An early diagnosis and treatment of IE might avoid its spread to a second valve.

## 1. Introduction

Infective endocarditis (IE) is a severe condition associated with high mortality and frequent complications [1]. Multivalvular IE (MIE) is relatively uncommon, with incidence ranging from 12% to 30% [2,3]. MIE information is scarce, and most published data come from case reports and surgical treatment techniques [3,4,5,6,7,8,9,10]. Few studies have compared MIE with single-valve IE (SIE) [2,9,11]. The prognostic influence of MIE is unclear as some data suggest an association with poor outcomes [3,4,5,12,13,14,15], whereas others do not [2,6,7,8,9,11]. The aim of our study was to describe the profile and prognosis of patients with MIE and to compare them with SIE in a large cohort of IE patients.

## 2. Materials and Methods

The Spanish Collaboration on Endocarditis, *Grupo de Apoyo al Manejo de la endocarditis infecciosa en ESpaña* (GAMES), is a national observational prospective registry that has been previously described [16,17,18,19]. Multidisciplinary teams, including infectious disease physicians, cardiologists, cardiac surgeons, microbiologists, echocardiographers, and other imaging specialists, have completed standardized case report forms with information regarding IE episodes and follow-up data. A complete list of GAMES members is shown in Appendix A. IE patients at 38 Spanish hospitals between January 2008 and 2020 were included. Inclusion criteria were the diagnosis of definite valvular IE by modified Duke criteria [20]. IE management, including the decision to perform surgery and the type of surgery, was performed by the local medical team following the 2009 and 2015 European Society of Cardiology recommendations [1]. MIE was considered present when two or more valves were involved. Valve involvement was defined by echocardiography as valves with vegetations or new regurgitation or by direct intraoperative visualization of vegetations.

This study complied with the principles outlined in the Declaration of Helsinki and was approved by the ethics committee of participating centers.

### Statistical Methods

Continuous variables are summarized as means ± standard deviations (SDs) or medians and interquartile ranges when normal distribution was not observed, as per the Kolmogorov–Smirnov goodness-of-fit test; categorical variables are expressed as numbers and percentages. Student‘s *t*-test, Mann–Whitney U test, or paired *t*-test was used to compare the continuous variables. The categorical variables were compared using the χ^2^ test or Fisher’s exact test. Kaplan–Meier curves were used to assess the cumulative survival of patients with valvular IE according to the presence of SIE or MIE. The curves were compared with the log-rank test. Multivariable logistic regression analyses (backward selection) were performed to determine the mortality predictors and to assess the independent association of MIE with mortality. All variables with a *p*-value < 0.10 in univariate analyses were included in the multivariable analyses. The statistical analysis was performed using SPSS, version 22.0 (IBM, Armonk, NY, USA).

## 3. Results

Of 5900 patients with possible or definite IE, 4531 had definite IE, 4064 had definite valvular IE, and 577 with MIE (14.2%) (Figure 1).

In patients with MIE, the most common locations were mitral (552, 95.7%) and aortic (550, 95.3%). MIE was mainly due to mitral-aortic involvement (507, 87.9%), right or left IE was only seen in 68 patients (11.8%), and tricuspid-pulmonary involvement was rare (2, 0.3%). MIE involved only native valves in 450 patients (78.0%). Table 1 shows the baseline characteristics and clinical events of our patients with valvular IE.

Compared with patients with SIE, patients with MIE had a similar age (69 vs. 67 years), were more frequently men (67% vs. 73%), and had a more common streptococcal etiology (29% vs. 33%).

Complications were also more common in MIE, including intracardiac complications (36% vs. 50%) and heart failure (43% vs. 53%). MIE was associated with surgical indication (68% vs. 85%) and surgery (46% vs. 56%). Compared with patients with SIE, patients with MIE had higher in-hospital mortality (34% vs. 27%). MIE was an independent predictor of in-hospital mortality (odds ratio 1.3, 95% confidence interval 1.1–1.7) (Table 2A). One-year mortality was also higher in patients with MIE than in those with SIE (39% vs. 33%, Figure 2), although MIE did not have an independent association with one-year mortality (OR 1.1, 95% CI 0.9–1.4, *p* = 0.43) (Table 2B).

## 4. Discussion

In our large national cohort of valvular IE, one-seventh of patients had MIE, which was associated with a poor prognosis.

MIE can involve native and prosthetic valves. The primary event is bacterial adherence to damaged valves during bacteriemia, with later persistence and growth within the cardiac lesions causing local extensions and tissue damage [21]. MIE can be the result of a simultaneous infection in two previously damaged valves in a patient with persistent bacteriemia or, more often, sequential seeding of a previously damaged valve [11]. In other cases, the infection of the first valve creates a new valvular lesion. A jet of aortic regurgitation may damage and infect the anterior mitral leaflet, which can also cause mechanical complications in the leaflets and mitral valve apparatus [11,21,22]. Other proposed mechanisms are the formation of an anterior abscess spreading and the destruction of the mitral annulus, or the prolapsing aortic IE “kissing vegetation phenomenon” [23]. Bilateral IE is uncommon (12% in our series) and might be related to shunts produced by congenital heart diseases [3], intracardiac devices, or repeated injections by drug users.

The most common etiologies in our and other cohorts were streptococcal, staphylococcal, and enterococcal [2,3,4,5,6,7,8,9,11,24]. *S. viridans* infections more frequently occurred in MIE compared with SIE, a fact that has been previously reported [3,4,6,11,24]. However, the incidence of *Staphylococci* IE is increasing [25]. *S. gallolyticus* (*S. bovis*) also has a high frequency of multivalvular locations, especially in French registries [11], more often infecting advanced age patients with known heart disease.

Surgical treatment is frequently needed in IE and even more so in MIE. About 85% of our MIE patients had a surgical indication, a higher proportion than in SIE, although only in 56% of MIE patients was surgery finally performed. MIE patients develop heart failure more often than SIE patients, and heart failure is a common surgical indication [1]. In addition, patients with MIE frequently present extensive tissue destruction [24]. These two factors are probably related to the higher rate of surgical treatment in MIE than in SIE [3,4,6,9].

The prognostic influence of MIE is unclear. An association with poor outcomes has been described in some studies [3,4,5,12,13,14,15,26] but not in others [2,6,7,8,9,11,24]. Prosthetic IE is an important compounding factor when we compare MIE with SIE, as prosthetic IE is associated with a poor prognosis [4]. Table 3 summarizes the main information previously published regarding MIE. The mean age was higher in our cohort than in the published studies, probably because most previous data came from surgical series, and patients eligible for surgery are usually younger.

Comparisons of SIE and information regarding medically treated patients are scarce. Moreover, a surgical series of selected patients shows excellent results that do not reflect everyday clinical practice. Our series of MIE is the largest reported to date and includes both patients treated with and without surgery.

The limitations of this study should be noted. Local medical teams were responsible for IE management, including deciding on surgery, and any judgments may have been influenced by factors not registered in this study. In any case, our data come from a large national database and show a clear association of MIE with IE prognosis.

## 5. Conclusions

About one-seventh of the valvular IE patients had MIE, mainly due to mitral-aortic involvement. MIE is associated with a poor in-hospital prognosis. An early diagnosis and treatment of IE might avoid its spread to a second valve.

## Figures and Tables

**Figure 1 jcm-11-04736-f001:**
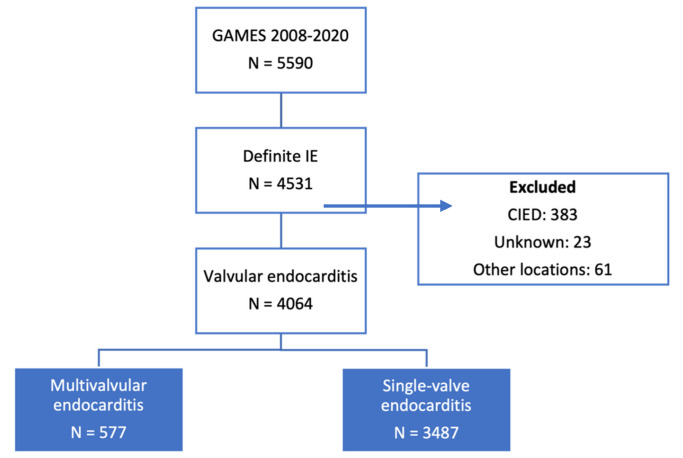
Flow chart of our cohort of patients from the Spanish Collaboration on Endocarditis, *Grupo de Apoyo al Manejo de la endocarditis infecciosa en ESpaña* (GAMES). IE: infective endocarditis. CIED: cardiac implantable electronic device.

**Figure 2 jcm-11-04736-f002:**
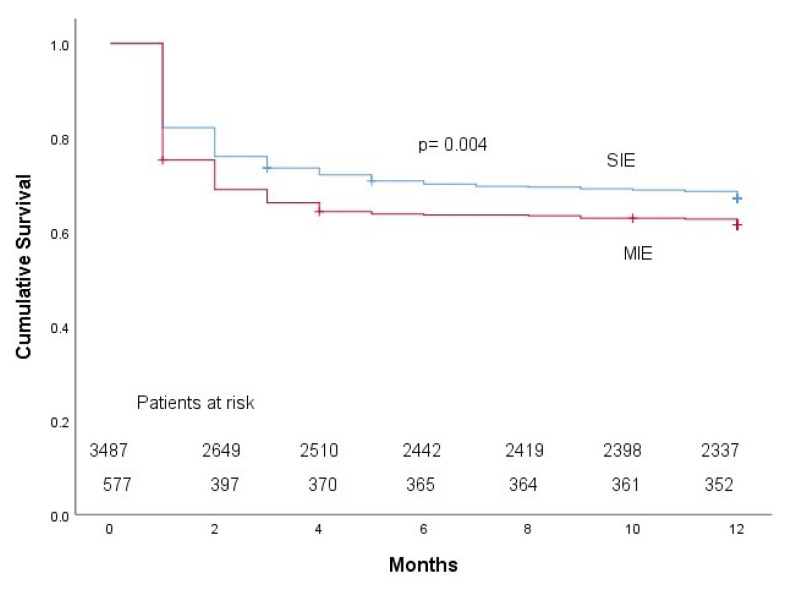
Kaplan–Meier curves for the cumulative survival of patients with valvular infective endocarditis (IE) according to the presence of single-valve IE (SIE) or multivalvular IE (MIE). *p*-value reflects the result of the log-rank test.

**Table 1 jcm-11-04736-t001:** Baseline characteristics and clinical events in patients with valvular infective endocarditis (IE) according to the presence of single-valve IE (SIE) or multivalvular IE (MIE).

	SIE (3487)	MIE (577)	*p*
Age, mean (IQR ^1^)	69 (57–77)	67 (57–76)	0.273
Sex (Men)	2341 (67.1%)	421 (72.9%)	0.005
Location			
Aortic	1827 (52.4%)	550 (95.3%)	<0.001
Mitral	1428 (41.0%)	552 (95.7%)	<0.001
Tricuspid	177 (5.1%)	57 (9.9%)	<0.001
Pulmonary	50 (1.4%)	18 (3.1%)	0.003
Native IE	2326 (66.7%)	450 (78.0%)	<0.001
Prosthetic IE	1169 (33.5%)	194 (33.6%)	0.963
Etiology			
*Staphylococcus aureus*	842 (24.1%)	113 (19.6%)	0.017
Coagulase-negative staphylococci	618 (17.7%)	94 (16.3%)	0.402
*Enterococcus*	541 (15.5%)	109 (18.9%)	0.040
*Streptococcus*	999 (28.6%)	192 (33.3%)	0.024
*Candida*	50 (1.4%)	7 (1.2%)	0.676
Clinical course			
Vegetation present	2759 (79.1%)	493 (85.4%)	<0.001
Intracardiac complications	1261 (36.2%)	291 (50.4%)	<0.001
Perforation or rupture	533 (15.2%)	150 (25.9%)	<0.001
Pseudoaneurysm	242 (6.9%)	56 (9.7%)	0.018
Abscess	682 (19.5%)	151 (26.1%)	0.001
Intracardiac fistula	98 (2.8%)	26 (4.5%)	0.028
Vascular phenomenon	378 (10.8%)	60 (10.4%)	0.751
New heart murmur	1261 (36.2%)	263 (45.6%)	<0.001
Heart failure	1489 (42.7%)	305 (52.9%)	<0.001
Persistent bacteriemia	404 (11.6%)	69 (12.0%)	0.796
Central nervous system involvement	777 (22.3%)	140 (24.3%)	0.292
Embolization	834 (23.9%)	153 (26.5%)	0.177
Renal failure	1278 (36.7%)	220 (38.1%)	0.495
Septic shock	465 (13.3%)	94 (16.3%)	0.056
Sepsis	648 (18.6%)	105 (18.2%)	0.825
Indication for surgery	2360 (67.7%)	491 (85.1%)	<0.001
Cardiac surgery	1616 (46.3%)	325 (56.3%)	<0.001
Surgery indicated not performed	772 (22.1%)	168 (29.1%)	<0.001
Mean hospital stay (IQR)	36 (22–52)	38 (22–54)	0.368
Antibiotic treatment days, mean (IQR)	40 (28–46)	38 (21–45)	0.368
In-hospital mortality	937 (26.9%)	198 (34.3%)	<0.001
1-year mortality	1146 (32.9%)	222 (38.5%)	0.008
IE Recurrence in those alive	42 (1.6%)	7 (1.8%)	0.777

^1^ IQR: interquartile range.

**Table 2 jcm-11-04736-t002:** A. Independent predictors of in-hospital mortality in patients with valvular infective endocarditis (IE). B. Independent predictors of 1-year mortality in patients with valvular infective endocarditis (IE).

**(A)**		
	**OR (95% CI)**	** *p* **
Male sex	0.8 (0.7–0.9)	0.041
Charlson comorbidity index	1.12 (1.09–1.16)	<0.001
Heart failure	2.9 (2.5–3.4)	<0.001
Multivalvular IE	1.3 (1.1–1.7)	0.004
Severe sepsis	2.1 (1.8–2.6)	<0.001
*S. aureus*	1.7 (1.4–2.1)	<0.001
Nosocomial IE	1.6 (1.3–1.9)	<0.001
Intracardiac abscess	1.3 (1.1–1.7)	0.004
Age (years)	1.015 (1.008–1.022)	<0.001
**(B)**		
	**OR (95% CI)**	** *p* **
Mitral location	1.2 (1.0–1.4)	0.017
Charlson comorbidity index	1.15 (1.12–1.19)	<0.001
Heart failure	2.5 (2.2–2.9)	<0.001
Persistent bacteriemia	1.2 (1.0–1.5)	0.043
Severe sepsis	2.0 (1.7–2.4)	<0.001
*S. aureus*	1.4 (1.2–1.7)	<0.001
Nosocomial IE	1.6 (1.3–1.9)	<0.001
Intracardiac abscess	1.4 (1.1–1.7)	<0.001
Age (years)	1.015 (1.015–1.009)	<0.001

**Table 3 jcm-11-04736-t003:** Previous data and our cohort results in patients with multivalvular infective endocarditis (MIE).

First Author, Year	N and Cohort Type	MIE	Mean/Median Age (Years)	Main Etiologies	MIE in-Hospital Mortality
*Kim et al., 2000* [2]	77	14 (31%)	65	*S. aureus* 43% *S. viridans* 36%	21% overall 29% surgery
*Mihalhevic et al., 2001* [4]	63 MIE surgery	All	49	*S. viridans* 28% *S. aureus*	16%
*Gillinov et al., 2001* [6]	54 MIE surgery	All	50	*Streptococci* 70% *Staphylococci* 18%	0%
*David et al., 2007* [7]	383 IE surgery	101 (26%)	51	*S. aureus* 23% *S. viridans* 18%	12%
*Yao et al., 2009* [3]	388 IE surgery	48 (12%)	42	*S. viridans* 29% *S. aureus* 19%	13%
*Sheikh et al., 2009* [8]	90 MIE surgery	All	53	*S. aureus* 16% *S. viridans* 14%	15%
*Selton et al., 2010* [11]	300 IE 511 IE	42 (14%) 88 (17%)	58 60	*Streptococci* 64% *S. aureus* 14%	26% 23%
*López et al., 2011* [15]	680 IE	115 (17%)	58	*S. aureus* 22% Coagulase-negative *staphylococci* 18%	30%
*Ota et al., 2011* [9]	152 native valve IE surgery	35 (23%)	47	*S. aureus* 22% *S. viridans* 22%	9%
*Kim et al., 2013* [24]	90 native valve IE surgery	23 (25%)	47	*S. viridans* 70%	0%
*Our cohort*	4064 IE	577 (14%)	67	*S. viridans* 33% *S. aureus* 20%	34%

## Data Availability

Data available under request from SEICAV.

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
