# Peer review of "Multivalvular Endocarditis: A Rare Condition with Poor Prognosis"

_jcm, 2022, doi:10.3390/jcm11164736_

Round 1

Reviewer 1 Report

With interest I have read the article Multivalvular endocarditis: a rare condition with poor prognosis by Alvarez-Zaballos and colleageaus. 

Is figure 2 correct? 

Could the authors describe what they did in the Methods as statistical analyses regarding the figure.

Why did the authors choose OR instead of HR for outcomes at 1-year (table 2B)?

Why did the authors not refer to the article of Holland and colleageaus (PMID: 31402124)?

Author Response

Reviewer 1:

With interest I have read the article Multivalvular endocarditis: a rare condition with poor prognosis by Alvarez-Zaballos and colleageaus. 

We would like to thank the comments from reviewer 1 that have helped us to improve our manuscript.

Is figure 2 correct? 

Could the authors describe what they did in the Methods as statistical analyses regarding the figure.

Figure 2 is correct and shows Kaplan-Meier curves for the cumulative survival of patients with valvular infective endocarditis according to the presence of single-valve infective endocarditis or multivalvular infective endocarditis. P-value reflects the result of the log-rank test. This has been clarified in the Methods and in the figure legend.

Why did the authors choose OR instead of HR for outcomes at 1-year (table 2B)?

The exact moment of death was unavailable in some patients. This is the reason why we used OR. In any case we have repeated the analyses with HR and the results are similar
Why did the authors not refer to the article of Holland and colleageaus (PMID: 31402124)?

The article of Holland et al is not focused on multivalvular endocarditis. In any case as the authors describe in a table that multivalvular involvement was associated with heart failure hospitalization we have included this reference.

Reviewer 2 Report

The chosen subject brings important clarifications related to the prognosis of patients with multivalvular infective endocarditis (MIE) compared to single-valve IE (SIE).

The large number of patients in the study group makes the conclusions relevant.

I believe that the article can be improved with the following aspects:

In line 144, reference is made to advanced age patients with known heart disease, but no details are given. In this group, we could also mention patients with congenital malformations that do not threaten life but can represent a cause of severe evolution in case of endocarditis.

There are recent articles in the literature, which show associations between endocarditis and various congenital malformations, as I exemplify below, and which would bring more information if the subject was explored in an additional sentence in the Discussions.

·       Leonard EJ, Kuebler BE, Zenni MM, Scuderi CB. A review of infective endocarditis associated with congenital heart diseaseConsultant. 2017;57(11):363-641.

·       Gug C, Gorduza EV, Lăcătuşu A, Vaida MA, Bîrsăşteanu F, Puiu M, Stoicănescu D. CHARGE syndrome associated with de novo (I1460Rfs*15) frameshift mutation of CHD7 gene in a patient with arteria lusoria and horseshoe kidney. Exp Ther Med. 2020 Jul;20(1):479-485. doi: 10.3892/etm.2020.8683. Epub 2020 Apr 23. PMID: 32509017; PMCID: PMC7271729.

Here it could be mentioned how patients with cardiac implantable electronic devices or diabetic patients evolve; there are well-chosen citations in this sense (18, 19).

In line 151, these aspects are mentioned, but too succinctly: An association with poor outcome has 151 been described in some studies [3, 4, 5, 12, 13, 14, 15] but not in others [2, 6, 7, 8, 9, 11, 24].

In line 156, Table 3. draws attention to the comparison between the study groups reported by different authors. Thus it becomes visible that the average age of 67 years of the current cohort is the highest. A comment related to this aspect should be made. Maybe the explanation is related to the age limits of the current group, or maybe other groups also had young or adolescent patients in the study group.

Author Response

Reviewer 2:

The chosen subject brings important clarifications related to the prognosis of patients with multivalvular infective endocarditis (MIE) compared to single-valve IE (SIE).

The large number of patients in the study group makes the conclusions relevant.

We would like to thank the comments from reviewer 2 that have helped us to improve our manuscript.

I believe that the article can be improved with the following aspects:

In line 144, reference is made to advanced age patients with known heart disease, but no details are given. In this group, we could also mention patients with congenital malformations that do not threaten life but can represent a cause of severe evolution in case of endocarditis.

There are recent articles in the literature, which show associations between endocarditis and various congenital malformations, as I exemplify below, and which would bring more information if the subject was explored in an additional sentence in the Discussions.

  • Leonard EJ, Kuebler BE, Zenni MM, Scuderi CB. A review of infective endocarditis associated with congenital heart disease. Consultant. 2017;57(11):363-641.
  • Gug C, Gorduza EV, Lăcătuşu A, Vaida MA, Bîrsăşteanu F, Puiu M, Stoicănescu D. CHARGE syndrome associated with de novo (I1460Rfs*15) frameshift mutation of CHD7 gene in a patient with arteria lusoria and horseshoe kidney. Exp Ther Med. 2020 Jul;20(1):479-485. doi: 10.3892/etm.2020.8683. Epub 2020 Apr 23. PMID: 32509017; PMCID: PMC7271729.

Here it could be mentioned how patients with cardiac implantable electronic devices or diabetic patients evolve; there are well-chosen citations in this sense (18, 19).

      Please note that this manuscript is focused on multivalvular IE. We have published previous manuscripts of the same cohort regarding congenital heart disease, diabetes, and cardiac implantable electronic devices. Moreover, as explained in figure 1, patients with IE associated with cardiac implantable electronic devices were excluded from the present study.

In line 151, these aspects are mentioned, but too succinctly: An association with poor outcome has 151 been described in some studies [3, 4, 5, 12, 13, 14, 15] but not in others [2, 6, 7, 8, 9, 11, 24].

In line 156, Table 3. draws attention to the comparison between the study groups reported by different authors. Thus it becomes visible that the average age of 67 years of the current cohort is the highest. A comment related to this aspect should be made. Maybe the explanation is related to the age limits of the current group, or maybe other groups also had young or adolescent patients in the study group.

The mean age was higher in our cohort than in published studies, probably because most previous data come from surgical series, and patients eligible for surgery are usually younger. We have clarified this in the discussion.